# Risk of hospitalization and mortality due to COVID-19 in people with obesity: An analysis of data from a Brazilian state

Erika Cardoso dos Reis[1]*, Phillipe Rodrigues[2], Tatielle Rocha de Jesus[3], Elma Lúcia de Freitas Monteiro[4], Jair Sindra Virtuoso Junior[4], Lucas Bianchi[5]

1 Department of Clinical and Social Nutrition, School of Nutrition, Federal University of Ouro Preto, Ouro Preto, Minas Gerais, Brazil, 2 School of Physical Education and Sports, Federal University of Rio de Janeiro, Rio de Janeiro, Rio de Janeiro, Brazil, 3 Department of Integrated Health Education, Federal University of Espírito Santo, Vitória, Espírito Santo, Brazil, 4 Graduate Program in Health Care, Federal University of Triângulo Mineiro, Uberaba, Minas Gerais, Brazil, 5 National School of Public Health (ENSP/Fiocruz), Rio de Janeiro, Rio de Janeiro, Brazil

* erika.careis@gmail.com

**Data Availability Statement:** All relevant data from this study are third party data and are available from the database: https://coronavirus.es.gov.br/

## Abstract

The aim of this article is to assess the odds ratio of hospitalization and mortality due to COVID-19 in people with obesity using data from residents of Espírito Santo, Brazil. An observational, quantitative, cross-sectional study was carried out from the database available on the official channel of the State Health Secretariat of Espírito Santo. Crude odds ratio estimates (ORs) referring to the association between variables were calculated, as well as adjusted odds ratios (adjusted odds ratios—OR adj.) and their respective 95% confidence intervals (CI 95%). The results indicate that men, non-white, no education or with lower education level and age over 40 years old were more likely to be hospitalized and died of COVID-19. People with obesity are at risk of hospitalization and death due to COVID-19 54% and 113% higher than people who do not have obesity. People with obesity had a higher chance of hospitalization when they were over 40 years old, had breathing difficulty, and the comorbidities diabetes (2.18 higher) and kidney disease (4.10 higher). The odds ratio of death for people with obesity over 60 years old was 12.51 higher, and those who were hospitalized was 17.9 higher compared to those who were not hospitalized.

## Introduction

The hospitalization and mortality rates due to COVID-19 have varied considerably due to several aspects, such as age group, current comorbidities, socioeconomic conditions, among other characteristics [1–6]. Regarding comorbidities, chronic diseases such as diabetes, coronary heart -disease and obesity have been associated with the worst prognosis for the disease [7–11].

When the first studies on risk factors for the severity of the disease began to be published, obesity was identified as one of those in which the risk of hospitalization and death increased,

painel-covid-19-es. The authors confirm that they did not have any special access privileges.

**Funding:** The project received financial support from the Pan American Health Organization (PAHO).

**Competing interests:** NO authors have competing interests.

which throughout the pandemic period was confirmed by different systematic reviews [12–14]. However, few published studies that investigate the role of obesity as a risk factor for the severity of COVID-19 were carried out in Brazil. Carneiro et al (2021) investigated the relationship between overweight and obesity with the COVID-19 mortality rate in Brazilian states [15]. The authors found a positive and significant correlation between the variables. Souza et al (2021) in a study carried out with information on notified cases of the disease, noticed that individuals with heart disease, diabetes and declining age present a worse health outcome; in addition, they identified that socioeconomic conditions would also be associated with a worse outcome. Thus, they conclude that COVID-19 affects different population groups differently and unequally [16].

Thus, considering the high prevalence of people with overweight and obesity in Brazil, added to the still out-of-control pandemic context, it is important to know the factors related to hospitalization and death in people with obesity, to establish protection mechanisms for this population.

In this context, this cross-sectional study aims to assess associated odds ratio of hospitalization and mortality due to COVID-19 in people with obesity based on data from Espírito Santo residents, Brazil.

## Method

This is an observational, quantitative, cross-sectional study, conducted from the database available on the official channel of the Health Department of Espírito Santo Government, "COVID-19 Panel", for the dissemination of coronavirus cases in state level (https://coronavirus.es.gov.br/painel-COVID-19-es). The COVID-19 Panel is a system developed by government and powered by the eSUS/Health Surveillance System (eSUS/VS), which records all suspected and/or confirmed cases of COVID-19 in the state of Espírito Santo (ESPÍRITO SANTO, 2020) from notification forms filled out by health professionals from health units throughout the state.

This study included all patients confirmed by COVID-19 in Espírito Santo, until September 10th, 2020, which corresponded to 118,138 cases, according to Fig 1. The confirmation and notification of cases followed the criteria of Technical Note COVID-19 No. 29/2020 –GEVS/SESA/ES, elaborated by Health Department of Espírito Santo: 1. Case confirmed by laboratory diagnosis: the positive result Reverse Transcription—Polymerase Chain Reaction (RT-PCR) in real time per validated protocol; or the positive validated serological test (rapid test). 2. Case confirmed by clinical-epidemiological diagnosis: suspected case with a history of close or home contact with a laboratory confirmed case for COVID-19 [17]. Individuals with a confirmed diagnosis for COVID-19 and who had the evolution of the case closed (cure or death by COVID-19) were selected for this study. Therefore, all those who were still undergoing treatment for the disease, without information or who died of other causes, were excluded.

The study variables were derived from the eSUS/VS System notification forms, considering the following patient data: age group, gender, race/color, education level, signs and symptoms (defined by the database the options: fever, breathing difficulty, cough, running nose, sore throat, diarrhea and headache), comorbidities (defined by the database the options: lung disease, cardiovascular disease, kidney disease, diabetes, smoking, obesity), hospitalization (yes / no) and evolution (cure / death by COVID-19). This study was based on the STROBE guidelines for reporting observational studies [18].

Considering the study design, cross-sectional, the study population was observed only once and information regarding the outcome and exposure was collected at the same time. Thus, this study is a Fig 1 of the population and the associations which were noticed here do not

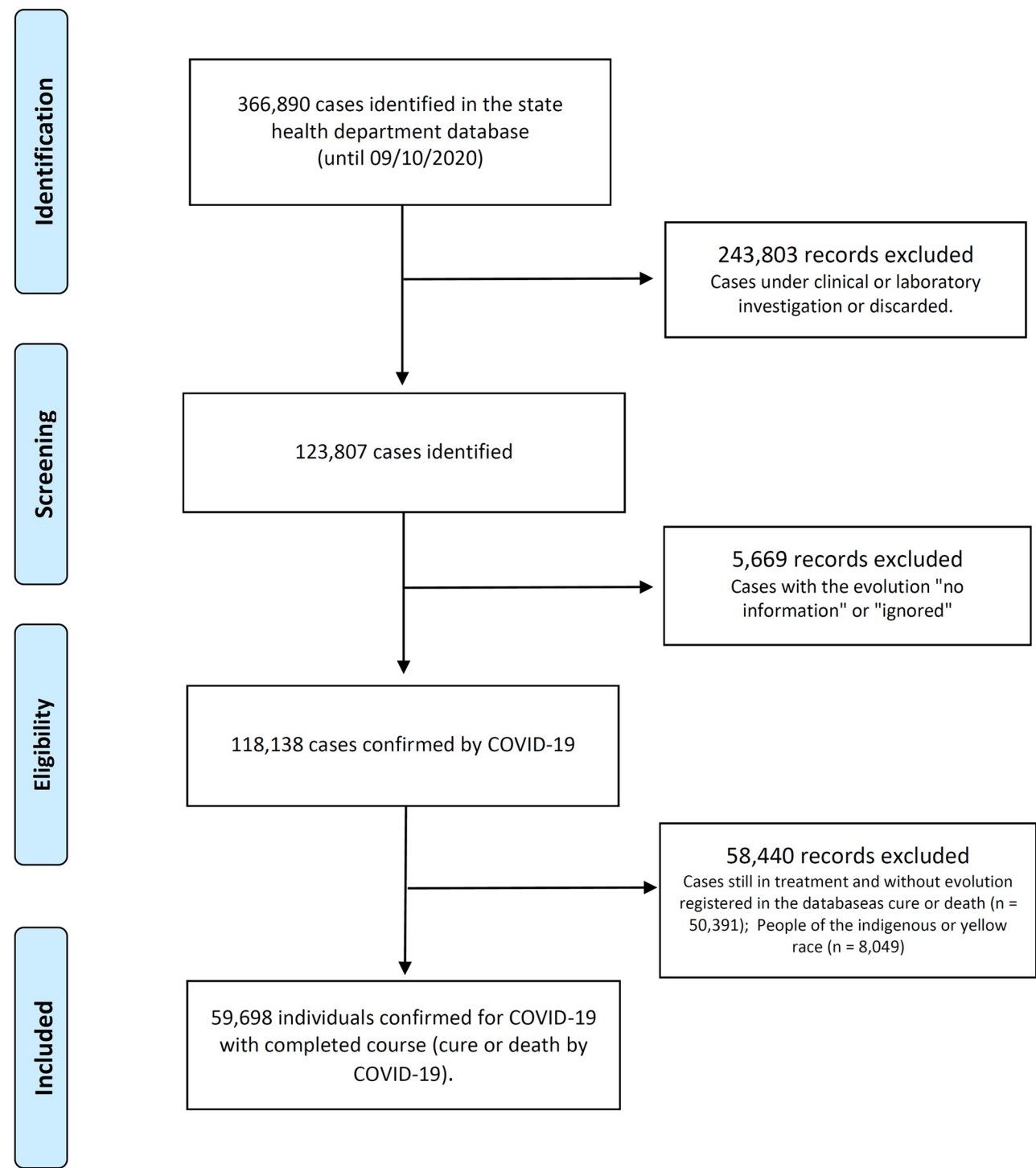

**Fig 1. Flow diagram of the selection of individuals participating in the study.**

have a cause-effect relationship. To quantify the noticed associations, crude odds ratio estimates (*odds ratio*—OR) were presented for the association among exposure variables and the outcome, as well as adjusted odds ratios (adjusted *odds ratio*—OR adj.) and their respective 95% confidence intervals (95%CI).

All analyzes were performed using R.4.0.3 software.

The study was carried out in accordance with the ethical principles of Resolution 466/2012 of the National Health Council; the approval of the work required by the Research Ethics Committee was not necessary, due to the use of secondary data, with free access and without identification of the subjects.

## Results

Since the beginning of the pandemic until September 10[th], 2020, 118,138 cases were confirmed by COVID-19 in the state of Espírito Santo. People who had not yet evolved (cure or death by COVID-19) described in the database were excluded, and data from 59.698 people were analyzed, and of these, 3025 were people with obesity (Fig 1).

Tables 1 and 2 show the characteristics of people with COVID-19 considered in this study by hospitalized group and mortality.

Most people who were hospitalized because of COVID-19 were over 60 years old (59.8%), male (41.8%), black or brown (58.0%) and had full high school (23.6%).

Regarding signs and symptoms, 68.5% had cough, 68.5% fever and 29.5% headache. The most frequent comorbidities among hospitalized patients were cardiovascular disease (55.6%), diabetes (30.9%) and obesity (10.3%).

Among people who died of COVID-19, most people were male (57.8%), were black or brown (54.3%), over 60 years old (76.7%), and had incomplete elementary school (4 to 8 years of schooling) (17.8%). Regarding signs and symptoms, cough and fever were the most frequent symptoms, 65.7% and 62.2% respectively. The most common registered comorbidities were cardiovascular disease (60.0%) and diabetes (34.1%), followed by obesity (10.9%).

According to the results presented in Table 3, fixing the other variables, it was identified that women are 37% less likely to be hospitalized by COVID-19 when compared to men. Black individuals are 21% more likely to be hospitalized by COVID-19 than white individuals.

Individuals from 40 to 59 years old and 60 years old or more had a chance of hospitalization by COVID-19 2.05 and 6.11 times, respectively, the chance of people from 18 to 39 years old.

Regarding education level, no education people and those with incomplete elementary school have a chance of hospitalization by COVID-19 respectively, 1.85 and 1.90 times the chance of people with a university diploma to be hospitalized.

Obesity, fever and breathing difficulty are characteristics associated with the chance of hospitalization by COVID-19. When compared to people who do not have these characteristics, the chance of hospitalization is, respectively, 1.62, 1.46 and 5.80 times in symptomatic cases. Running nose, sore throat, diarrhea and headache presented values indicating a "protective effect" for hospitalization by COVID-19, that is, in cases which individuals had these symptoms, there was a reduction in the chance of hospitalization by 45%, 48%, 32% and 60% compared to individuals who did not have these symptoms.

Individuals who have heart and kidney diseases, diabetes and smoke have increased chances of hospitalization by COVID-19, respectively, 1.38, 2.56, 1.71 and 1.97 times the chance of those who do not have these conditions of health.

The data in Table 4 only consider people with obesity, fixing the other variables, and indicate that women have 40% less chance of being hospitalized by COVID-19 when compared to men; black individuals are 61% more likely to be hospitalized by COVID-19 than white individuals to be hospitalized; people from 40 to 59 years old and 60 years old or more reflect a chance of hospitalization by COVID-19 1,98 and 4,23 times, respectively, the chance of people from 18 to 39 years old to be hospitalized. Education level does not impact the chance of hospitalization by COVID-19 in this scenario.

**Table 1. Sociodemographic characteristics, comorbidities and signs and symptoms of confirmed and hospitalized cases with COVID-19, Espírito Santo, Brazil, 2020.**

| Variables | Hospitalized | | | | | |
|---|---|---|---|---|---|---|
| | Yes | | No | | Not informed | |
| | n = 1110 | % (CI 95%) | n = 34586 | % (CI 95%) | n = 24002 | % (CI 95%) |
| **Gender** | | | | | | |
| Male | 646 | 58.2% (55.3–61.1) | 14575 | 42.1% (41.6–42.7) | 10744 | 44.8% (44.1–45.4) |
| Female | 464 | 41.8% (38.9–44.7) | 20011 | 57.9% (57.3–58.4) | 13258 | 55.2% (54.6–55.9) |
| **Race/Color:** | | | | | | |
| White | 466 | 42.0% (39.1–44.9) | 16069 | 46.5% (45.9–47.0) | 11418 | 47.6% (46.9–48.2) |
| Black/Brown | 644 | 58.0% (55.1–60.9) | 18517 | 53.5% (53.0–54.1) | 12584 | 52.4% (51.8–53.1) |
| **Age Group** | | | | | | |
| 18 to 39 years old | 125 | 11.3% (9.5–13.3) | 16172 | 46.8% (46.2–47.3) | 11720 | 48.8% (48.2–49.5) |
| 40 to 59 years old | 321 | 28.9% (26.3–31.7) | 13264 | 38.4% (37.8–38.9) | 8892 | 37.0% (36.4–37.7) |
| 60 years old or more | 664 | 59.8% (56.9–62.7) | 5150 | 14.9% (14.5–15.3) | 3390 | 14.1% (13.7–14.6) |
| **Education level** | | | | | | |
| No education | 103 | 9.3% (7.7–11.1) | 565 | 1.6% (1.5–1.8) | 350 | 1.5% (1.3–1.6) |
| Incomplete elementary school | 232 | 20.9% (18.6–23.4) | 2866 | 8.3% (8.0–8.6) | 2923 | 12.2% (11.8–12.6) |
| Full elementary school | 192 | 17.3% (15.2–19.6) | 4657 | 13.5% (13.1–13.8) | 3169 | 13.2% (12.8–13.6) |
| Incomplete primary school | 140 | 12.6% (10.8–14.7) | 1876 | 5.4% (5.2–5.7) | 1142 | 4.8% (4.5–5.0) |
| Full primary school | 67 | 6.0% (4.8–7.6) | 1431 | 4.1% (3.9–4.4) | 1035 | 4.3% (4.1–4.6) |
| Full high school | 262 | 23.6% (21.2–26.2) | 15341 | 44.4% (43.8–44.9) | 10027 | 41.8% (41.2–42.4) |
| University diploma | 114 | 10.3% (8.6–12.2) | 7850 | 22.7% (22.3–23.1) | 5356 | 22.3% (21.8–22.8) |
| **Fever** | | | | | | |
| No | 392 | 35.3% (32.7–38.3) | 15686 | 45.4% (44.8–45.9) | 12592 | 52.5% (51.9–53.2) |
| Yes | 714 | 64.3% (61.7–67.3) | 18893 | 54.6% (54.1–55.2) | 11384 | 47.4% (46.8–48.1) |
| Missing | 4 (0.4%) | | 7 (0.0%) | | 26 (0.1%) | |
| **Breathing Difficulty** | | | | | | |
| No | 386 | 34.8% (32.1–37.7) | 26775 | 77.4% (77.0–77.9) | 19511 | 81.3% (80.9–81.9) |
| Yes | 722 | 65.0% (62.3–67.9) | 7805 | 22.6% (22.1–23.0) | 4466 | 18.6% (18.1–19.1) |
| Missing | 2 (0.2%) | | 6 (0.0%) | | 25 (0.1%) | |
| **Cough** | | | | | | |
| No | 345 | 31.1% (28.6–34.0) | 13107 | 37.9% (37.4–38.4) | 11083 | 46.2% (45.6–46.9) |
| Yes | 760 | 68.5% (66.0–71.4) | 21473 | 62.1% (61.6–62.6) | 12893 | 53.7% (53.1–54.4) |
| Missing | 5 (0.5%) | | 6 (0.0%) | | 26 (0.1%) | |
| **Running nose** | | | | | | |
| No | 873 | 78.6% (76.4–81.2) | 20148 | 58.3% (57.7–58.8) | 15455 | 64.4% (63.8–65.1) |
| Yes | 233 | 21.0% (18.8–23.6) | 14431 | 41.7% (41.2–42.3) | 8522 | 35.5% (34.9–36.2) |
| Missing | 4 (0.4%) | | 7 (0.0%) | | 25 (0.1%) | |
| **Sore throat** | | | | | | |
| No | 955 | 86.0% (84.1–88.2) | 22972 | 66.4% (65.9–66.9) | 16620 | 69.2% (68.7–69.9) |
| Yes | 152 | 13.7% (11.8–15.9) | 11606 | 33.6% (33.1–34.1) | 7357 | 30.7% (30.1–31.3) |
| Missing | 3 (0.3%) | | 8 (0.0%) | | 25 (0.1%) | |
| **Diarrhea** | | | | | | |
| No | 960 | 86.5% (84.7–88.7) | 27893 | 80.6% (80.2–81.1) | 19793 | 82.5% (82.1–83.0) |
| Yes | 146 | 13.2% (11.3–15.3) | 6686 | 19.3% (18.9–19.8) | 4184 | 17.4% (17.0–17.9) |
| Missing | 4 (0.4%) | | 7 (0.0%) | | 25 (0.1%) | |
| **Headache** | | | | | | |
| No | 778 | 70.1% (67.6–73.0) | 13603 | 39.3% (38.8–39.9) | 11476 | 47.8% (47.2–48.5) |

*(Continued)*

**Table 1.** (Continued)

| Variables | Hospitalized | | | | | |
|---|---|---|---|---|---|---|
| | **Yes** | | **No** | | **Not informed** | |
| | **n = 1110** | **% (CI 95%)** | **n = 34586** | **% (CI 95%)** | **n = 24002** | **% (CI 95%)** |
| Yes | 328 | 29.5% (27.0–32.4) | 20976 | 60.6% (60.1–61.2) | 12501 | 52.1% (51.5–52.8) |
| Missing | 4 (0.4%) | | 7 (0.0%) | | 25 (0.1%) | |
| **Obesity** | | | | | | |
| No | 992 | 89.4% (87.8–91.3) | 32814 | 94.9% (94.8–95.3) | 23037 | 96.0% (95.9–96.3) |
| Yes | 114 | 10.3% (8.7–12.2) | 1714 | 5.0% (4.7–5.2) | 934 | 3.9% (3.7–4.1) |
| Missing | 4 (0.4%) | | 58 (0.2%) | | 31 (0.1%) | |
| **Lung Disease** | | | | | | |
| No | 1014 | 91.4% (90.0–93.2) | 33308 | 96.3% (96.2–96.6) | 23280 | 97.0% (96.9–97.3) |
| Yes | 91 | 8.2% (6.8–10.0) | 1259 | 3.6% (3.4–3.8) | 696 | 2.9% (2.7–3.1) |
| Missing | 5 (0.5%) | | 19 (0.1%) | | 26 (0.1%) | |
| **Cardiovascular Disease** | | | | | | |
| No | 489 | 44.1% (41.3–47.2) | 26918 | 77.8% (77.4–78.3) | 20007 | 83.4% (83.0–83.9) |
| Yes | 617 | 55.6% (52.8–58.7) | 7652 | 22.1% (21.7–22.6) | 3970 | 16.5% (16.1–17.0) |
| Missing | 4 (0.4%) | | 16 (0.0%) | | 25 (0.1%) | |
| **Kidney Disease** | | | | | | |
| No | 1055 | 95.0% (94.0–96.5) | 34368 | 99.4% (99.3–99.5) | 23840 | 99.3% (99.3–99.5) |
| Yes | 51 | 4.6% (3.5–6.0) | 201 | 0.6% (0.5–0.7) | 137 | 0.6% (0.5–0.7) |
| Missing | 4 (0.4%) | | 17 (0.0%) | | 25 (0.1%) | |
| **Diabetes** | | | | | | |
| No | 763 | 68.7% (66.2–71.6) | 31782 | 91.9% (91.6–92.2) | 22576 | 94.1% (93.9–94.5) |
| Yes | 343 | 30.9% (28.4–33.8) | 2788 | 8.1% (7.8–8.4) | 1399 | 5.8% (5.5–6.1) |
| Missing | 4 (0.4%) | | 16 (0.0%) | | 27 (0.1%) | |
| **Smoking** | | | | | | |
| No | 1027 | 92.5% (91.3–94.3) | 33778 | 97.7% (97.5–97.9) | 23558 | 98.2% (98.1–98.4) |
| Yes | 78 | 7.0% (5.7–8.7) | 791 | 2.3% (2.1–2.5) | 417 | 1.7% (1.6–1.9) |
| Missing | 5 (0.5%) | | 17 (0.0%) | | 27 (0.1%) | |

Regarding the symptoms, breathing difficulty was a symptom associated with the chance of hospitalization due to COVID-19. When compared to people who do not have this symptom, the chance of hospitalization is, respectively, 3.20 times in symptomatic cases. Symptoms such as sore throat, diarrhea and headache presented values indicating a "protective effect" for hospitalization by COVID-19, that is, in cases which individuals had these symptoms, there was a reduction in the chance of hospitalization of 48%, 57% and 52% compared to individuals who did not have these symptoms.

Individuals who had kidney disease and diabetes have a chance of hospitalization by COVID-19, respectively, 5.32 and 2.04 times the chance of those who do not have these health problems.

Table 5 presents the logistic regression model for the association of sociodemographic factors and symptoms with death by COVID-19, and the data show that women are 34% less likely to evolve to death by COVID-19 when compared to men.

People from 40 to 59 years old or 60 years old or more have a chance of death by COVID-19 3.85 and 21.01 times, respectively, the chance of people from 18 to 39 years old of dying. Missing a university diploma is a risk factor for death by COVID-19, and the chance of death

**Table 2. Sociodemographic characteristics, comorbidities, and signs and symptoms of confirmed cases that died of COVID-19, Espírito Santo, Brazil, 2020.**

| Variables | Death by covid-19 | | | |
|---|---|---|---|---|
| | Yes | | No | |
| | N = 1406 | % (CI 95%) | N = 58292 | % (CI 95%) |
| **Gender** | | | | |
| Male | 813 | 57.8% (55.2–60.4) | 25152 | 43.1% (42.7–43.6) |
| Female | 593 | 42.2% (39.6–44.8) | 33140 | 56.9% (56.4–57.3) |
| **Race/Color:** | | | | |
| White | 643 | 45.7% (43.1–48.3) | 27310 | 46.9% (46.4–47.3) |
| Black | 763 | 54.3% (51.7–56.9) | 30982 | 53.1% (52.7–53.6) |
| **Age Group** | | | | |
| 18 to 39 years old | 52 | 3.7% (2.8–4.8) | 27965 | 48.0% (47.6–48.4) |
| 40 to 59 years old | 276 | 19.6% (17.6–21.8) | 22201 | 38.1% (37.7–38.5) |
| 60 years old or more | 1078 | 76.7% (74.4–78.8) | 8126 | 13.9% (13.7–14.2) |
| **Education level** | | | | |
| University diploma | 74 | 5.3% (4.2–6.6) | 13246 | 22.7% (22.4–23.1) |
| No education | 167 | 11.9% (10.3–13.7) | 851 | 1.5% (1.4–1.6) |
| Incomplete elementary school | 250 | 17.8% (15.9–19.9) | 5771 | 9.9% (9.7–10.1) |
| Full elementary school | 277 | 19.7% (17.7–21.9) | 7741 | 13.3% (13.0–13.6) |
| Incomplete primary school | 264 | 18.8% (16.8–20.9) | 2894 | 5.0% (4.8–5.1) |
| Full primary school | 117 | 8.3% (7.0–9.9) | 2416 | 4.1% (4.0–4.3) |
| Full high school | 257 | 18.3% (16.3–20.4) | 25373 | 43.5% (43.1–43.9) |
| **Fever** | | | | |
| No | 526 | 37.4% (35.0–40.1) | 28144 | 48.3% (47.9–48.7) |
| Yes | 875 | 62.2% (59.9–65.0) | 30116 | 51.7% (51.3–52.1) |
| Missing | 5 (0.4%) | | 32 (0.1%) | |
| **Breathing Difficulty** | | | | |
| No | 582 | 41.4% (38.9–44.0) | 46090 | 79.1% (78.8–79.4) |
| Yes | 822 | 58.5% (56.0–61.1) | 12171 | 20.9% (20.6–21.2) |
| Missing | 2 (0.1%) | | 31 (0.1%) | |
| **Cough** | | | | |
| No | 477 | 33.9% (31.6–36.6) | 24058 | 41.3% (40.9–41.7) |
| Yes | 924 | 65.7% (63.4–68.4) | 34202 | 58.7% (58.3–59.1) |
| Missing | 5 (0.4%) | | 32 (0.1%) | |
| **Running Nose** | | | | |
| No | 1112 | 79.1% (77.1–81.4) | 35364 | 60.7% (60.3–61.1) |
| Yes | 290 | 20.6% (18.6–22.9) | 22896 | 39.3% (38.9–39.7) |
| Missing | 4 (0.3%) | | 32 (0.1%) | |
| **Sore throat** | | | | |
| No | 1214 | 86.3% (84.8–88.3) | 39333 | 67.5% (67.1–67.9) |
| Yes | 187 | 13.3% (11.7–15.2) | 18928 | 32.5% (32.1–32.9) |
| Missing | 5 (0.4%) | | 31 (0.1%) | |
| **Diarrhea** | | | | |
| No | 1212 | 86.2% (84.7–88.3) | 47434 | 81.4% (81.1–81.7) |
| Yes | 188 | 13.4% (11.7–15.3) | 10828 | 18.6% (18.3–18.9) |
| Missing | 6 (0.4%) | | 30 (0.1%) | |
| **Headache** | | | | |
| No | 998 | 71.0% (68.9–73.6) | 24859 | 42.6% (42.3–43.1) |
| Yes | 401 | 28.5% (26.4–31.1) | 33404 | 57.3% (56.9–57.7) |

*(Continued)*

**Table 2.** (Continued)

| Variables | Death by covid-19 | | | |
|---|---|---|---|---|
| | Yes | | No | |
| | N = 1406 | % (CI 95%) | N = 58292 | % (CI 95%) |
| Missing | 7 (0.5%) | | 29 (0.0%) | |
| **Obesity** | | | | |
| No | 1246 | 88.6% (87.3–90.6) | 55597 | 95.4% (95.3–95.7) |
| Yes | 153 | 10.9% (9.4–12.7) | 2609 | 4.5% (4.3–4.7) |
| Missing | 7 (0.5%) | | 86 (0.1%) | |
| **Lung Disease** | | | | |
| No | 1280 | 91.0% (89.7–92.7) | 56322 | 96.6% (96.5–96.8) |
| Yes | 122 | 8.7% (7.3–10.3) | 1924 | 3.3% (3.2–3.5) |
| Missing | 4 (0.3%) | | 46 (0.1%) | |
| **Cardiovascular Disease** | | | | |
| No | 557 | 39.6% (37.2–42.3) | 46857 | 80.4% (80.1–80.8) |
| Yes | 844 | 60.0% (57.7–62.8) | 11395 | 19.5% (19.2–19.9) |
| Missing | 5 (0.4%) | | 40 (0.1%) | |
| **Kidney Disease** | | | | |
| No | 1325 | 94.2% (93.2–95.6) | 57938 | 99.4% (99.4–99.5) |
| Yes | 77 | 5.5% (4.4–6.8) | 312 | 0.5% (0.5–0.6) |
| Missing | 4 (0.3%) | | 42 (0.1%) | |
| **Diabetes** | | | | |
| No | 921 | 65.5% (63.2–68.2) | 54200 | 93.0% (92.8–93.3) |
| Yes | 480 | 34.1% (31.8–36.8) | 4050 | 6.9% (6.7–7.2) |
| Missing | 5 (0.4%) | | 42 (0.1%) | |
| **Smoking** | | | | |
| No | 1314 | 93.5% (92.5–95.0) | 57049 | 97.9% (97.8–98.1) |
| Yes | 86 | 6.1% (5.0–7.5) | 1200 | 2.1% (1.9–2.2) |
| Missing | 6 (0.4%) | | 43 (0.1%) | |
| **Hospitalized** | | | | |
| No | 514 | 36.6% (34.1–39.1) | 34072 | 58.5% (58.0–58.9) |
| Not informed | 305 | 21.7% (19.6–23.9) | 23697 | 40.7% (40.3–41.1) |
| Yes | 587 | 41.7% (39.2–44.3) | 523 | 0.9% (0.8–1.0) |

may range from 1.61 to 4.08 times, and the behavior of decreasing OR is almost linear as education level increases.

Obesity increased the chance of death by 2.08 times when compared to people who did not have this condition. Symptoms such as fever and breathing difficulty were associated with the chance of death by COVID-19. When compared to people who do not have these symptoms, the chance of dying is, respectively, 1.43 and 3.15 times in symptomatic cases. Running nose, sore throat, diarrhea and headache presented values that indicate a "protective effect" for death by COVID-19, that is, in cases which individuals had these symptoms, there was a reduction in the chance of death of 32%, 37%, 21% and 48% compared to individuals who did not have these symptoms.

Individuals who had lung, heart, kidney diseases and diabetes have a chance of dying by COVID-19, respectively, 1.34, 1.24, 2.90 and 1.70 times the chance of those who do not have these health problems.

**Table 3. Logistic regression model for the association of sociodemographic factors and symptoms with hospitalization for COVID-19, Espírito Santo, Brazil, 2020.**

| Variables | n | OR | CI 95% | Logistic Model | |
|---|---|---|---|---|---|
| | | | | OR adj. | CI 95% |
| **Gender** | | | | | |
| Male | 25905 | 1,00 | Ref. | 1,00 | Ref. |
| Female | 33670 | 0,53 | 0,47; 0,60*** | 0,63 | 0,55; 0,73*** |
| **Race/Color** | | | | | |
| White | 27899 | 1,00 | Ref. | 1,00 | Ref. |
| Black | 31676 | 1,21 | 1,07; 1,36** | 1,21 | 1,05; 1,38** |
| **Age (Notification date)** | | | | | |
| 18 to 39 years old | 27958 | 1,00 | Ref. | 1,00 | Ref. |
| 40 to 59 years old | 22434 | 3,08 | 2,51; 3,81*** | 2,05 | 1,65; 2,56*** |
| 60 years old or more | 9183 | 16,52 | 13,66; 20,14*** | 6,11 | 4,86; 7,73*** |
| **Education level** | | | | | |
| University diploma | 13293 | 1,00 | Ref. | 1,00 | Ref. |
| No education | 1015 | 12,68 | 9,57; 16,78*** | 1,85 | 1,34; 2,55*** |
| Incomplete elementary school | 6015 | 5,54 | 4,42; 6,99*** | 1,90 | 1,48; 2,46*** |
| Full elementary school | 8007 | 2,82 | 2,23; 3,58*** | 1,27 | 0,99; 1,65 |
| Incomplete primary school | 3151 | 5,10 | 3,96; 6,59*** | 1,08 | 0,81; 1,43 |
| Full primary school | 2531 | 3,25 | 2,38; 4,41*** | 0,95 | 0,67; 1,33 |
| Full high school | 25563 | 1,18 | 0,95; 1,48 | 0,95 | 0,75; 1,20 |
| **Obesity** | | | | | |
| No | 56814 | 1,00 | Ref. | 1,00 | Ref. |
| Yes | 2761 | 2,21 | 1,80; 2,69*** | 1,62 | 1,28; 2,04*** |
| **Fever** | | | | | |
| No | 28634 | 1,00 | Ref. | 1,00 | Ref. |
| Yes | 30941 | 1,52 | 1,34; 1,72*** | 1,46 | 1,27; 1,68*** |
| **Breathing difficulty** | | | | | |
| No | 46613 | 1,00 | Ref. | 1,00 | Ref. |
| Yes | 12962 | 6,37 | 5,61; 7,23*** | 5,80 | 5,06; 6,66*** |
| **Cough** | | | | | |
| No | 24505 | 1,00 | Ref. | 1,00 | Ref. |
| Yes | 35070 | 1,35 | 1,18; 1,53** | 1,06 | 0,92; 1,23 |
| **Running nose** | | | | | |
| No | 36425 | 1,00 | Ref. | 1,00 | Ref. |
| Yes | 23150 | 0,37 | 0,32; 0,43*** | 0,55 | 0,47; 0,65*** |
| **Sore throat** | | | | | |
| No | 40475 | 1,00 | Ref. | 1,00 | Ref. |
| Yes | 19100 | 0,31 | 0,26; 0,37*** | 0,52 | 0,43; 0,62*** |
| **Diarrhea** | | | | | |
| No | 48569 | 1,00 | Ref. | 1,00 | Ref. |
| Yes | 11006 | 0,63 | 0,53; 0,75*** | 0,68 | 0,56; 0,82*** |
| **Headache** | | | | | |
| No | 25809 | 1,00 | Ref. | 1,00 | Ref. |
| Yes | 33766 | 0,28 | 0,24; 0,31*** | 0,40 | 0,34; 0,46*** |
| **Lung Disease** | | | | | |
| No | 57531 | 1,00 | Ref. | 1,00 | Ref. |
| Yes | 2044 | 2,39 | 1,90; 2,96*** | 1,25 | 0,96; 1,62 |
| **Heart Disease** | | | | | |

*(Continued)*

**Table 3.** (Continued)

| Variables | n | OR | CI 95% | Logistic Model | |
|---|---|---|---|---|---|
| | | | | OR adj. | CI 95% |
| No | 47353 | 1,00 | Ref. | 1,00 | Ref. |
| Yes | 12222 | 4,43 | 3,93; 5,01*** | 1,38 | 1,19; 1,61*** |
| **Kidney Disease** | | | | | |
| No | 59189 | 1,00 | Ref. | 1,00 | Ref. |
| Yes | 386 | 8,39 | 6,07; 11,39*** | 2,56 | 1,75; 3,71*** |
| **Diabetes** | | | | | |
| No | 55054 | 1,00 | Ref. | 1,00 | Ref. |
| Yes | 4521 | 5,15 | 4,51; 5,89*** | 1,71 | 1,45; 2,00*** |
| **Smoking** | | | | | |
| No | 58291 | 1,00 | Ref. | 1,00 | Ref. |
| Yes | 1284 | 3,27 | 2,55; 4,13*** | 1,97 | 1,47; 2,60*** |

**Abbreviations:** OR—*odds ratio*; 95% CI; 95% confidence interval.

*** p-value <0.001;

** 0.001 ≤ p-value <0.01;

* 0.01 ≤-p-value <0.05.

n: Number of individuals with the exposure who presented the outcome.

The results presented in Table 6 only consider people with obesity and show that women are 32% less likely to evolve to death by COVID-19 when compared to men. People from 40 to 59 years old or 60 years old or more increase the chance of death by COVID-19 in 3.86 and 20.28 times, respectively, when compared to the chance of people from 18 to 39 years old to die. Missing a university diploma is a risk factor for death by COVID-19, the chance of death may range from 1.60 to 3.99 times.

Fever and breathing difficulty are symptoms associated with the chance of death by COVID-19. When compared to people who do not have these symptoms, the chance of dying is, respectively, 1.43 and 3.22 times in symptomatic cases. Running nose, sore throat, diarrhea and headache showed estimates that indicate a "protective effect" for death by COVID-19, that is, in cases which individuals had this symptom, there was a reduction in the chance of death in 33%, 37%, 21% and 47% compared to individuals who did not present these symptoms.

Individuals with kidney disease and diabetes have a chance of dying by COVID-19 of 5.32 and 2.04 times, respectively, the chance of individuals who do not have these comorbidities of dying. It is important to emphasize that the low prevalence of death in patients with kidney disease, may have resulted in its statistical significance.

## Discussion

Data showed that men, non-white, no education or with low education level and declining age were more likely to be hospitalized and die of COVID-19 in the state of Espírito Santo.

The severity of the disease according to gender has also been assessed in other studies [19,20]. Previous research has shown that the X chromosome is known to keep the largest number of genes related to the immune system in the entire genome. Women, for presenting chromosome XX, are generally more responsive to infections [19]. In addition, studies show that in males there is a greater presence of receptors for SARS-CoV-2, the Angiotensin-Converting Enzyme 2 (ECA2), in their alveolar cells if compared to women [21].

**Table 4. Logistic regression model for the association of sociodemographic factors and symptoms with hospitalization by COVID-19 in patients with obesity, Espírito Santo, Brazil, 2020.**

| Variables | n | OR | CI 95% | Logistic Model | |
|---|---|---|---|---|---|
| | | | | OR adj. | CI 95% |
| **Gender** | | | | | |
| Male | 25921 | 1,00 | Ref. | 1,00 | Ref. |
| Female | 33701 | 0,59 | 0,40; 0,87*** | 0,60 | 0,39; 0,93** |
| **Race/Color** | | | | | |
| White | 27914 | 1,00 | Ref. | 1,00 | Ref. |
| Black | 31708 | 1,24 | 0,84; 1,83 | 1,61 | 1,04; 2,50*** |
| **Age (notification date)** | | | | | |
| 18 to 39 years old | 27980 | 1,00 | Ref. | 1,00 | Ref. |
| 40 to 59 years old | 22454 | 2,66 | 1,54; 4,80*** | 1,98 | 1,09; 3,75*** |
| 60 years old or more | 9188 | 7,41 | 4,29; 13,43*** | 4,23 | 2,15; 8,59*** |
| **Education level** | | | | | |
| University diploma | 13300 | 1,00 | Ref. | 1,00 | Ref. |
| No education | 1015 | 3,17 | 1,09; 8,13*** | 0,96 | 0,29; 2,83 |
| Incomplete elementary school | 6018 | 1,59 | 0,77; 3,22 | 0,72 | 0,32; 1,62 |
| Full elementary school | 8011 | 1,38 | 0,71; 2,71 | 0,87 | 0,41; 1,83 |
| Incomplete primary school | 3151 | 2,21 | 0,97; 4,62 | 0,79 | 0,33; 1,87 |
| Full primary school | 2531 | 2,34 | 0,98; 5,18 | 1,29 | 0,50; 3,16 |
| Full high school | 25596 | 0,84 | 0,47; 1,54 | 0,87 | 0,47; 1,68 |
| **Fever** | | | | | |
| No | 28655 | 1,00 | Ref. | 1,00 | Ref. |
| Yes | 30967 | 1,15 | 0,78; 1,74 | 1,26 | 0,80; 2,02 |
| **Breathing difficulty** | | | | | |
| No | 46648 | 1,00 | Ref. | 1,00 | Ref. |
| Yes | 12974 | 3,56 | 2,40; 5,36*** | 3,20 | 2,09; 4,97*** |
| **Cough** | | | | | |
| No | 24520 | 1,00 | Ref. | 1,00 | Ref. |
| Yes | 35102 | 1,22 | 0,79; 1,95 | 1,45 | 0,88; 2,47 |
| **Running nose** | | | | | |
| No | 36450 | 1,00 | Ref. | 1,00 | Ref. |
| Yes | 23172 | 0,46 | 0,30; 0,70*** | 0,74 | 0,45; 1,19 |
| **Sore throat** | | | | | |
| No | 40519 | 1,00 | Ref. | 1,00 | Ref. |
| Yes | 19103 | 0,34 | 0,20; 0,55*** | 0,52 | 0,30; 0,88*** |
| **Diarrhea** | | | | | |
| No | 48612 | 1,00 | Ref. | 1,00 | Ref. |
| Yes | 11010 | 0,36 | 0,19; 0,63*** | 0,43 | 0,22; 0,77*** |
| **Headache** | | | | | |
| No | 25837 | 1,00 | Ref. | 1,00 | Ref. |
| Yes | 33785 | 0,34 | 0,23; 0,51*** | 0,48 | 0,31; 0,74*** |
| **Lung Disease** | | | | | |
| No | 57578 | 1,00 | Ref. | 1,00 | Ref. |
| Yes | 2044 | 1,59 | 0,78; 2,92 | 1,00 | 0,43; 2,10 |
| **Heart Disease** | | | | | |
| No | 47391 | 1,00 | Ref. | 1,00 | Ref. |
| Yes | 12231 | 2,16 | 1,46; 3,25*** | 1,02 | 0,64; 1,65 |

*(Continued)*

**Table 4.** (Continued)

| Variables | n | OR | CI 95% | Logistic Model | |
|---|---|---|---|---|---|
| | | | | OR adj. | CI 95% |
| **Kidney Disease** | | | | | |
| No | 59235 | 1,00 | Ref. | 1,00 | Ref. |
| Yes | 387 | 7,94 | 2,96; 19,52*** | 5,32 | 1,63; 16,23*** |
| **Diabetes** | | | | | |
| No | 55096 | 1,00 | Ref. | 1,00 | Ref. |
| Yes | 4526 | 3,34 | 2,25; 4,92*** | 2,04 | 1,29; 3,21*** |
| **Smoking** | | | | | |
| No | 58338 | 1,00 | Ref. | 1,00 | Ref. |
| Yes | 1284 | 3,07 | 1,58; 5,56*** | 2,16 | 0,98; 4,46 |

Abbreviations: OR—*odds ratio*; 95% CI; 95% confidence interval.

*** p-value <0.001;

** $0.001 \leq$ p-value <0.01;

* $0.01 \leq$-p-value <0.05.

n: Number of individuals with the exposure who presented the outcome.

Takahashi et al. (2020) while monitoring 98 patients with COVID-19 admitted to Yale Hospital from March 18[th] to May 9[th], 2020, noticed significantly higher levels of pro-inflammatory chemokines and cytokines in male participants, such as IL-8, IL-18 and CCL5, and a significantly lower number of T cells, both in the total count and in the proportion of live cells, over the course of the disease, which contributed to the worsening of their clinical condition [22].

Other authors also evaluated the frequency of race/color in people with COVID-19 and identified differences. An analysis carried out in the United Kingdom noticed that hospitalization by COVID-19 was found in 32 out of 7714 (0.4%) black participants, 28 out of 10.614 (0.2%) Asian participants and 489 out of 400,438 (0.1%) white participants [23]. A similar result was noticed in a study conducted in Detroit, United States, in which 2.316 (63.7%) people diagnosed with COVID-19 and who were hospitalized, 55.7% were black/brown [24].

Analyzes by Baqui et al (2020) with 11.321 Brazilian patients diagnosed with COVID-19 showed that, after age, the most important factor for hospital mortality was being brown or, to a lesser extent, black compared to white race [25].

Racial differences in the frequency of aggravation of COVID-19 can be multifactorial and are still unclear. These data may reflect differences in working conditions and health determinants they are submitted to, as well as being related to potential biological factors [23,26,27]. However, black/brown Brazilians have, on average, less economic security, live in favorable conditions to contagion, are less likely to be able to work remotely and constitute a substantial proportion of health workers, making them the most vulnerable to COVID– 19 [28].

In our study, no education people or those with lower education level had a higher chance of hospitalization and death, which can be explained by less access to information and health services, possibly having the incomes affected during the pandemic and living in inadequate hygienic and sanitary conditions [29].

In the investigation of 45,161 questionnaires carried out nationwide, by Oswaldo Cruz Foundation (Fiocruz), it was highlighted that the groups that least adhered to the social distance initiatives to control COVID-19 were composed by men (31.7%), from 30 to 49 years old (36.4%), with low education level (33.0%) and who kept working during the pandemic (81.3%) [30].

**Table 5. Logistic regression model for the association of sociodemographic factors and symptoms with death by COVID-19, Espírito Santo, Brazil, 2020.**

| Variables | n | OR | CI 95% | Logistic Model | |
|---|---|---|---|---|---|
| | | | | OR adj. | CI 95% |
| **Gender** | | | | | |
| Male | 25905 | 1,00 | Ref. | 1,00 | Ref. |
| Female | 33670 | 0,55 | 0,50; 0,62*** | 0,66 | 0,58; 0,76*** |
| **Race/Color** | | | | | |
| White | 27899 | 1,00 | Ref. | 1,00 | Ref. |
| Black | 31676 | 1,05 | 0,94; 1,16 | 1,00 | 0,87; 1,14 |
| **Age (Notification date)** | | | | | |
| 18 to 39 years old | 27958 | 1,00 | Ref. | 1,00 | Ref. |
| 40 to 59 years old | 22434 | 6,66 | 5,00; 9,06*** | 3,85 | 2,85; 5,31*** |
| 60 years old or more | 9183 | 70,33 | 53,77; 94,15*** | 21,01 | 15,61; 28,84*** |
| **Education level** | | | | | |
| University diploma | 13293 | 1,00 | Ref. | 1,00 | Ref. |
| No education | 1015 | 34,68 | 26,24; 46,24*** | 4,08 | 2,90; 5,77*** |
| Incomplete elementary school | 6015 | 7,62 | 5,90; 9,96*** | 2,23 | 1,64; 3,06*** |
| Full elementary school | 8007 | 6,33 | 4,92; 8,25*** | 2,88 | 2,15; 3,91*** |
| Incomplete primary school | 3151 | 16,07 | 12,44; 21,00*** | 2,94 | 2,17; 4,03*** |
| Full primary school | 2531 | 8,66 | 6,47; 11,66*** | 2,29 | 1,63; 3,25*** |
| Full high school | 25563 | 1,80 | 1,40; 2,35*** | 1,61 | 1,21; 2,18*** |
| **Obesity** | | | | | |
| No | 56814 | 1,00 | Ref. | 1,00 | Ref. |
| Yes | 2761 | 2,63 | 2,21; 3,12*** | 2,08 | 1,66; 2,59*** |
| **Fever** | | | | | |
| No | 28634 | 1,00 | Ref. | 1,00 | Ref. |
| Yes | 30941 | 1,55 | 1,39; 1,73*** | 1,43 | 1,24; 1,64*** |
| **Breathing difficulty** | | | | | |
| No | 46613 | 1,00 | Ref. | 1,00 | Ref. |
| Yes | 12962 | 5,35 | 4,80; 5,96*** | 3,15 | 2,75; 3,61*** |
| **Cough** | | | | | |
| No | 24505 | 1,00 | Ref. | 1,00 | Ref. |
| Yes | 35070 | 1,35 | 1,21; 1,51*** | 0,96 | 0,83; 1,11 |
| **Running nose** | | | | | |
| No | 36425 | 1,00 | Ref. | 1,00 | Ref. |
| Yes | 23150 | 0,40 | 0,35; 0,46*** | 0,68 | 0,58; 0,79*** |
| **Sore throat** | | | | | |
| No | 40475 | 1,00 | Ref. | 1,00 | Ref. |
| Yes | 19100 | 0,32 | 0,27; 0,37*** | 0,63 | 0,52; 0,75*** |
| **Diarrhea** | | | | | |
| No | 48569 | 1,00 | Ref. | 1,00 | Ref. |
| Yes | 11006 | 0,67 | 0,57; 0,78*** | 0,79 | 0,66; 0,96** |
| **Headache** | | | | | |
| No | 25809 | 1,00 | Ref. | 1,00 | Ref. |
| Yes | 33766 | 0,30 | 0,26; 0,33*** | 0,52 | 0,45; 0,60*** |
| **Lung Disease** | | | | | |
| No | 57531 | 1,00 | Ref. | 1,00 | Ref. |
| Yes | 2044 | 2,76 | 2,27; 3,33*** | 1,34 | 1,02; 1,73** |
| **Heart Disease** | | | | | |

*(Continued)*

**Table 5.** (Continued)

| Variables | n | OR | CI 95% | Logistic Model | |
|---|---|---|---|---|---|
| | | | | OR adj. | CI 95% |
| No | 47353 | 1,00 | Ref. | 1,00 | Ref. |
| Yes | 12222 | 6,25 | 5,60; 6,97*** | 1,24 | 1,07; 1,43*** |
| **Kidney Disease** | | | | | |
| No | 59189 | 1,00 | Ref. | 1,00 | Ref. |
| Yes | 386 | 10,42 | 7,99; 13,43*** | 2,90 | 2,00; 4,13*** |
| **Diabetes** | | | | | |
| No | 55054 | 1,00 | Ref. | 1,00 | Ref. |
| Yes | 4521 | 6,99 | 6,22; 7,84*** | 1,70 | 1,46; 1,98*** |
| **Smoking** | | | | | |
| No | 58291 | 1,00 | Ref. | 1,00 | Ref. |
| Yes | 1284 | 3,09 | 2,45; 3,86*** | 1,21 | 0,88; 1,65 |
| **Hospitalized** | | | | | |
| No | 34511 | 1,00 | Ref. | 1,00 | Ref. |
| Not informed | 23965 | 0,86 | 0,74; 1,02 | 0,96 | 0,82; 1,12 |
| Yes | 1099 | 75,23 | 64,95; 87,22*** | 19,59 | 16,41; 23,42*** |

**Abbreviations:** OR—*odds ratio*; 95% CI; 95% confidence interval.

*** p-value <0.001;

** 0.001 ≤ p-value <0.01;

* 0.01 ≤-p-value <0.05.

n: Number of individuals with the exposure who presented the outcome.

Regarding signs and symptoms, cough, headache and fever were the most ordinary identified ones in our study. Fever and breathing difficulty increased the chances of hospitalization and death, while running nose, sore throat, diarrhea and headache were shown to be protective effects. In addition, the fact of being hospitalized increased the chances of death in almost 20 times. In Wuhan, China, the most ordinary symptoms at the beginning of the disease in 138 hospitalized people were fever (136 [98.6%]), fatigue (96 [69.6%]), dry cough (82 [59.4%]), myalgia (48 [34.8%]) and dyspnea (43 [31.2%]). The less ordinary symptoms were headache, dizziness, abdominal pain, diarrhea, nausea and vomiting [31].

For 278 positive patients for COVID-19 in New York, the presence of gastrointestinal symptoms was associated with a longer duration of the disease, however, with a tendency for a lower rate of admission to the Intensive Care Unit and lower mortality [32].

In our analyses have also shown that obesity, heart, kidney and lung diseases, diabetes and smoking increased the chances of hospitalization. Obesity represented 4.5% of the total diagnoses of COVID-19, among comorbidities, it was the third risk factor that most increased the chances of hospitalization and the second related to the increase of the chances of death.

Vardavas and Nikitara (2020) evidenced in their systematic review that smoking patients were more likely to worsen COVID-19 than non-smokers [33]. Smoking is related to a higher expression of SARS-CoV-2 receptors, which can be the reason for the highest prevalence of more severe symptoms in this subgroup of patients [34]. In the study by Azar et al [35], comorbidities such as congestive heart failure or type 2 diabetes were associated with a greater chance of hospitalization compared to those who did not have these conditions.

Bello-Chavolla et al. [36] when evaluating the confirmed and negative cases of COVID-19 and their demographic and health characteristics in the General Directorate of Epidemiology

**Table 6.** Logistic regression model for the association of sociodemographic factors and symptoms with death by COVID-19 in patients with obesity, Espírito Santo, Brazil, 2020.

| Variables | n | OR | CI 95% | Logistic Model | |
|---|---|---|---|---|---|
| | | | | OR adj. | CI 95% |
| **Gender** | | | | | |
| Male | 25921 | 1,00 | Ref. | 1,00 | Ref. |
| Female | 33701 | 0,55 | 0,50; 0,62*** | 0,68 | 0,59; 0,78*** |
| **Race/Color** | | | | | |
| White | 27914 | 1,00 | Ref. | 1,00 | Ref. |
| Black | 31708 | 1,05 | 0,94; 1,16 | 0,99 | 0,86; 1,13 |
| **Age (Notification date)** | | | | | |
| 18 to 39 years old | 27980 | 1,00 | Ref. | 1,00 | Ref. |
| 40 to 59 years old | 22454 | 6,66 | 5,00; 9,06*** | 3,86 | 2,86; 5,31*** |
| 60 years old or more | 9188 | 70,33 | 53,77; 94,15*** | 20,28 | 15,07; 27,84*** |
| **Education level** | | | | | |
| University diploma | 13300 | 1,00 | Ref. | 1,00 | Ref. |
| No education | 1015 | 34,68 | 26,24; 46,24*** | 3,99 | 2,84; 5,64*** |
| Incomplete elementary school | 6018 | 7,62 | 5,90; 9,96*** | 2,19 | 1,61; 3,01*** |
| Full elementary school | 8011 | 6,33 | 4,92; 8,25*** | 2,86 | 2,13; 3,87*** |
| Incomplete primary school | 3151 | 16,07 | 12,44; 21,00*** | 2,91 | 2,15; 3,99*** |
| Full primary school | 2531 | 8,66 | 6,47; 11,66*** | 2,27 | 1,61; 3,21*** |
| Full high school | 25596 | 1,80 | 1,40; 2,35*** | 1,60 | 1,20; 2,16*** |
| **Fever** | | | | | |
| No | 28655 | 1,00 | Ref. | 1,00 | Ref. |
| Yes | 30967 | 1,55 | 1,39; 1,73*** | 1,43 | 1,24; 1,64*** |
| **Breathing difficulty** | | | | | |
| No | 46648 | 1,00 | Ref. | 1,00 | Ref. |
| Yes | 12974 | 5,35 | 4,80; 5,96*** | 3,22 | 2,81; 3,68*** |
| **Cough** | | | | | |
| No | 24520 | 1,00 | Ref. | 1,00 | Ref. |
| Yes | 35102 | 1,35 | 1,21; 1,51*** | 0,96 | 0,83; 1,11 |
| **Running nose** | | | | | |
| No | 36450 | 1,00 | Ref. | 1,00 | Ref. |
| Yes | 23172 | 0,40 | 0,35; 0,46*** | 0,67 | 0,57; 0,79*** |
| **Sore throat** | | | | | |
| No | 40519 | 1,00 | Ref. | 1,00 | Ref. |
| Yes | 19103 | 0,32 | 0,27; 0,37*** | 0,63 | 0,52; 0,75*** |
| **Diarrhea** | | | | | |
| No | 48612 | 1,00 | Ref. | 1,00 | Ref. |
| Yes | 11010 | 0,67 | 0,57; 0,78*** | 0,79 | 0,66; 0,95*** |
| **Headache** | | | | | |
| No | 25837 | 1,00 | Ref. | 1,00 | Ref. |
| Yes | 33785 | 0,30 | 0,26; 0,33*** | 0,53 | 0,46; 0,61*** |
| **Lung Disease** | | | | | |
| No | 57578 | 1,00 | Ref. | 1,00 | Ref. |
| Yes | 2044 | 2,76 | 2,27; 3,33*** | 1,35 | 1,04; 1,75** |
| **Heart Disease** | | | | | |
| No | 47391 | 1,00 | Ref. | 1,00 | Ref. |
| Yes | 12231 | 6,25 | 5,60; 6,97*** | 1,29 | 1,12; 1,50*** |

*(Continued)*

**Table 6.** (Continued)

| Variables | n | OR | CI 95% | Logistic Model | |
|---|---|---|---|---|---|
| | | | | OR adj. | CI 95% |
| **Kidney Disease** | | | | | |
| No | 59235 | 1,00 | Ref. | 1,00 | Ref. |
| Yes | 387 | 10,42 | 7,99; 13,43*** | 2,87 | 1,98; 4,09*** |
| **Diabetes** | | | | | |
| No | 55096 | 1,00 | Ref. | 1,00 | Ref. |
| Yes | 4526 | 6,99 | 6,22; 7,84*** | 1,76 | 1,51; 2,05*** |
| **Smoking** | | | | | |
| No | 58338 | 1,00 | Ref. | 1,00 | Ref. |
| Yes | 1284 | 3,09 | 2,45; 3,86*** | 1,27 | 0,92; 1,72 |
| **Hospitalized** | | | | | |
| No | 34552 | 1,00 | Ref. | 1,00 | Ref. |
| Yes | 1100 | 75,23 | 64,95; 87,22*** | 19,70 | 16,50; 23,55*** |
| Not informed | 23970 | 0,86 | 0,74; 0,99 | 0,96 | 0,83; 1,12 |

**Abbreviations:** OR—*odds ratio*; 95% CI; 95% confidence interval.

*** p-value <0.001;

** $0.001 \leq$ p-value <0.01;

* $0.01 \leq$ -p-value <0.05.

n: Number of individuals with the exposure who presented the outcome.

of the Ministry of Health of Mexico found that 51,633 individuals tested positive for SARS-CoV-2. When assessing age, there was a reduced chance of positivity for SARS-CoV-2 in patients <40 years old. However, in stratified models, it was found that for patients with diabetes, positivity for SARS-CoV-2 was associated with obesity, male gender and age <40 years old. Patients with obesity who had COVID-19 confirmed had an almost five-time increase in the risk of mortality (OR = 4.989; 95% CI = 4.444–5.600). In addition, they also had higher rates of Intensive Care Unit admission (5.0% vs. 3.3%) and were more likely to be intubated (5.2% vs. 3.3%) [36].

In our study, the highest chances of hospitalization and death for people with obesity were related to age over 60 years old, followed by the age group from 40 to 59 years old, who had breathing difficulties, diabetes and who had been hospitalized. The analysis by Klang et al. (2020), in New York City, with data from 3,406 patients, 572 patients under 50 years old and 2.834 over 50 years old have shown that in the youngest age group, 60 (10.5%) patients died, and the analysis univariate demonstrated that, for the youngest group, BMI $\geq$ 40 kg / m$^2$ was significantly associated with death (p <0.001) [37].

In the research by Ong et al. (2020), in Singapore, in patients under 60 years old it was verified that BMI $\geq$25kg / m$^2$ was significantly associated with pneumonia on chest X-ray at admission (p = 0.017), requiring low oxygen supplementation flow (OR = 6.32; 95% CI = 1.23–32.34) and mechanical ventilation (OR = 1.16; 95% CI = 1.00–1.34). BMI $\geq$25kg / m$^2$ was also associated with significantly higher serum levels of lactate dehydrogenase (p = 0.011), which were associated with the severity of the disease [38].

The mechanisms involving the role of obesity in the pathogenesis of COVID-19 are not yet well defined, but individuals with obesity generally have a decreased immune response to infectious pathogens, which can also affect the lung parenchyma, increasing the risk of inflammatory lung diseases [39]. In addition, as it is characterized as a low-grade inflammation, in

obesity mononuclear cells increase the transcription of pro-inflammatory cytokines, which increases the secretion of these cytokines [40].

Evidence suggests that adipose tissue is a pro-immunogenic and richly vascularized organ, with the ability to increase the pro-inflammatory response to viral infection. Thus, it can potentiate and prolong viral shedding in an environment that is already inflamed with the local amplification of cytokines, which can hinder the patient's recovery [41].

Zhang and colleagues in a logistic regression model also identified the factors that address the mechanisms underlying obesity predisposing COVID-19 patients to death. Through which the index related to inflammation, PCR, heart damage (hs-cTnI and *NT-proBNP*) and increased clotting activity (D-dimer) are characterized as significantly associated with adverse clinical outcomes in patients with high BMI. In addition, the decrease in lymphocytes and eosinophils or in total globulin levels was also correlated with the poor prognosis in these patients [42].

Abdominal obesity can restrict ventilation, preventing diaphragm excursion, as it reduces the compliance of the lung, chest wall and the entire respiratory system, resulting in decreased blood oxygen saturation and breathing functional capability [41,43].

Therefore, the data analyzed here confirm those found by other authors and show that obesity can be considered a risk factor for hospitalization and death by COVID-19, especially when in addition to obesity, other conditions such as age over 40 years old are present (more severe for those over 60 years old), the presence of comorbidities such as diabetes and kidney disease.

## Study limitations

This study has some limitations that deserve to be highlighted. Among them are the limitations of a cross-sectional study that analyzed data from a specific time and did not assess other aspects related to the illness of these individuals over time. Another important limitation is that this study is based on data from the state health department, which are obtained through the records in the Health Units, and although the notification forms have a lot of mandatory registration information, one cannot be sure about the recording quality of these data.

## Author Contributions

**Conceptualization:** Erika Cardoso dos Reis.

**Data curation:** Erika Cardoso dos Reis, Elma Lúcia de Freitas Monteiro, Jair Sindra Virtuoso Junior, Lucas Bianchi.

**Formal analysis:** Erika Cardoso dos Reis, Phillipe Rodrigues, Tatielle Rocha de Jesus, Lucas Bianchi.

**Methodology:** Elma Lúcia de Freitas Monteiro, Jair Sindra Virtuoso Junior.

**Validation:** Elma Lúcia de Freitas Monteiro, Jair Sindra Virtuoso Junior.

**Writing – original draft:** Erika Cardoso dos Reis, Phillipe Rodrigues, Tatielle Rocha de Jesus.

**Writing – review & editing:** Erika Cardoso dos Reis, Phillipe Rodrigues, Tatielle Rocha de Jesus, Elma Lúcia de Freitas Monteiro, Jair Sindra Virtuoso Junior, Lucas Bianchi.

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
