## [Decision Letter · Decision Letter 0]

6 Jul 2021

PONE-D-21-10679

Risk of hospitalization and mortality due to COVID-19 in people with obesity: An analysis of data from a Brazilian state

PLOS ONE

Dear Dr. Reis,

Thank you for submitting your manuscript to PLOS ONE. After careful consideration, we feel that it has merit but does not fully meet PLOS ONE’s publication criteria as it currently stands. Therefore, we invite you to submit a revised version of the manuscript that addresses the points raised during the review process.

We look forward to receiving your revised manuscript.

Kind regards,

Jennifer A. Hirst, DPhil

Academic Editor

PLOS ONE

Additional Editor Comments:

Please change yellow to “other non-white ethnicity”

Change illiterate to no education

Typo – page 17 “Individuals who presented lang, heart, kidney diseases and diabetes” should read “Individuals who presented lung, heart, kidney diseases and diabetes”

Journal Requirements:

Reviewers' comments:

Reviewer's Responses to Questions

**Comments to the Author**

1. Is the manuscript technically sound, and do the data support the conclusions?

Reviewer #1: No

Reviewer #2: No

2. Has the statistical analysis been performed appropriately and rigorously? 

Reviewer #1: No

Reviewer #2: No

3. Have the authors made all data underlying the findings in their manuscript fully available?

Reviewer #1: Yes

Reviewer #2: Yes

4. Is the manuscript presented in an intelligible fashion and written in standard English?

Reviewer #1: No

Reviewer #2: No

5. Review Comments to the Author

Reviewer #1: The authors report on the association between hospitalization/death and an array of factors (demographics, comorbidity and symptoms/signs) for patients with COVID-19 as residents of Espiroto Santo Brazil.

General:

Please adhere to the STROBE guidelines for reporting of observational studies: https://www.equator-network.org/reporting-guidelines/strobe/

Introduction

1) Please explain why your study is needed. It seems that the question your study tries to answer is already answered.

2) Regarding the objective: “… this study aims to assess the risk of hospitalization and mortality

due to COVID-19 in people with obesity based on data from Espírito Santo residents, Brazil.”

The study methods and results imply a more wide approach. Please adjust the aims to be consistent with the rest of the manuscript.

Methods

3) How are patients includes in this database, what were the selection criteria? How many were eligible and how many were not included?

4) Describe methods of follow-up? How long was the follow-up?

5) How was race/ ethnicity determined? Self reported? Determined by physician? What do you mean by yellow ect?

6) Regarding death, there maybe patients remaining in hospital that have no yet reached an outcome (either death or discharged). This means there is a high risk of (non)-differential misclassification which could lead to bias.

7) Are the authors using a causal or a prognostic approach? Presently, the “adjusted model” is not suitable for causal interpretation nor is it adequately validated for use as a prognostic model. Please see also these references below:

doi: 10.1097/EDE.0000000000001258

DOI: 10.1097/EDE.0000000000001259

8) A such, there also seems to be a Table 2 fallacy problem with the analyses:

DOI: 10.1093/aje/kws412

Results

9) Please report, according to STROBE item 13: “numbers of individuals at each stage of study—eg numbers potentially eligible, examined for eligibility, confirmed eligible, included in the study, completing follow-up, and analysed” A flow diagram is highly recommended.

10) How many patients had missing data and how was this handled?

11) Why present OR when you can present RR?

12) Table 2-5: It would be helpful to add the number of patients to each table, e.g. n of x men were hospitalized and n of y women were hospitalized.

Discussion

13) The discussion could benefit from a thorough limitations section.

Reviewer #2: Dear Editor,

Cardioso dos Reis and colleagues performed a cross-sectional study attempting to assess predictors and risk factors for hospitalization and death among patients with obesity and COVID-19 in Brazil.

The study has very important data from underreported patient population; therefore, it deserves special attention. On the other hand, I have to mention that this manuscript needs a lot of work in several aspects including English language editing, clear description of the methods, and organized and consistent presentation of the results and findings.

Below you can see some comments.

Introduction

*I would make the introduction much shorter. In specific, I would remove the first three paragraphs since the general information provided there is well-known. I would start directly with risk factors/obesity.

Methods

*This section needs English language editing.

*Authors need to make sure they report all component of STROBE checklist.

*Inclusion and exclusion criteria can be defined more clearly.

*How was obesity defined? Did researchers rely on documentation or they used BMI or other tool?

Results

*This section needs English language editing, as well.

*Would change obese patients to patients with obesity

*Table 1: What is yellow color/race?

*Do the rest of the tables present univariate or multivariate associations? This should be clear in the text and the table notes.

*Why some variables were placed in the regression, while others were not?

6. PLOS authors have the option to publish the peer review history of their article (what does this mean?). If published, this will include your full peer review and any attached files.

Reviewer #1: No

Reviewer #2: No

---

## [Author Response · Author response to Decision Letter 0]

31 Aug 2021

Dear Editor,

We greatly appreciate the reading and suggestions made in our manuscript “Risk of hospitalization and mortality due to COVID-19 in people with obesity: An analysis of data from a Brazilian state”. The manuscript was modified according to the reviewers' suggestions and we answer the questions presented point by point.

---

## [Editor Report · Decision Letter 1]

4 Sep 2021

PONE-D-21-10679R1

Risk of hospitalization and mortality due to COVID-19 in people with obesity: An analysis of data from a Brazilian state

PLOS ONE

Dear Dr. Reis,

Thank you for submitting your manuscript to PLOS ONE. After careful consideration, we feel that it has merit but does not fully meet PLOS ONE’s publication criteria as it currently stands. Therefore, we invite you to submit a revised version of the manuscript that addresses the points raised during the review process.

We look forward to receiving your revised manuscript.

Kind regards,

Jennifer A. Hirst, DPhil

Academic Editor

PLOS ONE

Journal Requirements:

Additional Editor Comments (if provided):

Please can the authors upload a marked-up version of the manuscript to allow reviewers to view changes that have been made from the previous version
---

## [Author Response · Author response to Decision Letter 1]

13 Sep 2021

Dear editor,

I hope you are all right. I send the correct documents.

Thank you,

Regards,

Dra. Erika Reis

---

## [Decision Letter · Decision Letter 2]

25 Nov 2021

PONE-D-21-10679R2Risk of hospitalization and mortality due to COVID-19 in people with obesity: An analysis of data from a Brazilian statePLOS ONE

Dear Dr. Reis,

Thank you for submitting your manuscript to PLOS ONE. After careful consideration, we feel that it has merit but does not fully meet PLOS ONE’s publication criteria as it currently stands. Therefore, we invite you to submit a revised version of the manuscript that addresses the points raised during the review process.

We look forward to receiving your revised manuscript.

Kind regards,

Jennifer A. Hirst, DPhil

Academic Editor

PLOS ONE

Journal Requirements:

Additional Editor Comments (if provided):

Please make the following changes to use more inclusive terminology:

Yellow is still used to describe Asian populations in the flow chart. Yellow is not a race or ethnicity, please change.

Please replace "illiterate" used throughout the manuscript with "No education"

Reviewers' comments:

Reviewer's Responses to Questions

**Comments to the Author**

1. If the authors have adequately addressed your comments raised in a previous round of review and you feel that this manuscript is now acceptable for publication, you may indicate that here to bypass the “Comments to the Author” section, enter your conflict of interest statement in the “Confidential to Editor” section, and submit your "Accept" recommendation.

Reviewer #1: (No Response)

2. Is the manuscript technically sound, and do the data support the conclusions?

Reviewer #1: Partly

3. Has the statistical analysis been performed appropriately and rigorously? 

Reviewer #1: Yes

4. Have the authors made all data underlying the findings in their manuscript fully available?

Reviewer #1: Yes

5. Is the manuscript presented in an intelligible fashion and written in standard English?

Reviewer #1: Yes

6. Review Comments to the Author

Reviewer #1: The authors have addressed most of my comments. However, the following issues require attention:

Comment 1:

Regarding Previous comment and response 6 (see below), the people who were still under treatment and who were excluded could have a different risk of dying than the included people and they can have a different distribution of co-variates. Hence there is a high risk of differential misclassification especially considering the large proportion of patients that were excluded (49%): 59.698 of 118.138 people were included.

Previous comment and response 6:

Regarding death, there may be patients remaining in hospital that have no yet reached an outcome (either death or discharged). This means there is a high risk of (non)- differential misclassification which could lead to bias.

No. We selected for this study people confirmed for COVID-19 who had the evolution of

the disease closed as cure or death. People who were still undergoing treatment were

excluded.

Comment 2:

Previous comment 7 (see below) has not been addressed by the authors.

Previous comment 7:

Are the authors using a causal or a prognostic approach? Presently, the “adjusted model” is not suitable for causal interpretation nor is it adequately validated for use as a prognostic model. Please see also these references below:

doi: 10.1097/EDE.0000000000001258

DOI: 10.1097/EDE.0000000000001259

Comment 3:

Regarding previous comment and response 8, the potential Table 2 fallacy problem has not been addressed by the authors (see below).

Previous comment and response 8:

A such, there also seems to be a Table 2 fallacy problem with the analyses:

DOI: 10.1093/aje/kws412

The symptoms of runny nose, sore throat, diarrhea and headache showed counterintuitive

associations, possibly due to confounding variables not properly controlled, resulting in

statistically significant associations, giving the false idea of a "protective effect" for both

hospitalization and death due to COVID-19.

7. PLOS authors have the option to publish the peer review history of their article (what does this mean?). If published, this will include your full peer review and any attached files.

Reviewer #1: No

---

## [Author Response · Author response to Decision Letter 2]

9 Jan 2022

Dear Editor

We greatly appreciate the new reading and suggestions made in our manuscript “Risk of hospitalization and mortality due to COVID-19 in people with obesity: An analysis of data from a Brazilian state”. The manuscript was modified according to the reviewers' suggestions and below we answer the questions presented point by point.

Below, we respond point by point to the comments and issues mentioned by the reviewers.

Reviewers' comments:

Please make the following changes to use more inclusive terminology:

Yellow is still used to describe Asian populations in the flow chart. Yellow is not a race or ethnicity, please change.

R: The term Yellow was changed to Asian populations.

Please replace "illiterate" used throughout the manuscript with "No education"

R: The term illiterate was changed to No education.

Comment 1:

Regarding Previous comment and response 6 (see below), the people who were still under treatment and who were excluded could have a different risk of dying than the included people and they can have a different distribution of co-variates. Hence there is a high risk of differential misclassification especially considering the large proportion of patients that were excluded (49%): 59.698 of 118.138 people were included.

Previous comment and response 6:

Regarding death, there may be patients remaining in hospital that have no yet reached an outcome (either death or discharged). This means there is a high risk of (non)- differential misclassification which could lead to bias. No. We selected for this study people confirmed for COVID-19 who had the evolution of the disease closed as cure or death. People who were still undergoing treatment were

excluded.

R: This is a cross-sectional study and as it is not intended to be a longitudinal study. For this reason, it cannot follow the evolution of the disease in individuals. Considering that individuals not included in this analysis (because their case was not closed) could have a higher risk of mortality, we would have to consider that these individuals could also have a lower risk of mortality and not go to the ICU. Because this is also a possibility and it cannot be considered a bias in the cross-sectional study.

Comment 2:

Previous comment 7 (see below) has not been addressed by the authors.

Previous comment 7:

Are the authors using a causal or a prognostic approach? Presently, the “adjusted model” is not suitable for causal interpretation nor is it adequately validated for use as a prognostic model. Please see also these references below:

doi: 10.1097/EDE.0000000000001258. 

DOI: 10.1097/EDE.0000000000001259.

R: Only association without causality and prognosis was measured.

Comment 3:

Regarding previous comment and response 8, the potential Table 2 fallacy problem has not been addressed by the authors (see below). 

Previous comment and response 8:

A such, there also seems to be a Table 2 fallacy problem with the analyses: DOI: 10.1093/aje/kws412 The symptoms of runny nose, sore throat, diarrhea and headache showed counterintuitive associations, possibly due to confounding variables not properly controlled, resulting in statistically significant associations, giving the false idea of a "protective effect" for both hospitalization and death due to COVID-19.

R: These are symptoms of mild COVID-19 and that is why in the study it has a protective effect. Because people who have these symptoms on hospitalization may have a lower chance of death.

King Regards,

Dr. Erika Cardoso dos Reis

---

## [Decision Letter · Decision Letter 3]

26 Jan 2022

Risk of hospitalization and mortality due to COVID-19 in people with obesity: An analysis of data from a Brazilian state

PONE-D-21-10679R3

Dear Dr. Reis,

We’re pleased to inform you that your manuscript has been judged scientifically suitable for publication and will be formally accepted for publication once it meets all outstanding technical requirements.

Kind regards,

Jennifer A. Hirst, DPhil

Academic Editor

PLOS ONE

Additional Editor Comments (optional):

Reviewers' comments:

Reviewer's Responses to Questions

**Comments to the Author**

1. If the authors have adequately addressed your comments raised in a previous round of review and you feel that this manuscript is now acceptable for publication, you may indicate that here to bypass the “Comments to the Author” section, enter your conflict of interest statement in the “Confidential to Editor” section, and submit your "Accept" recommendation.

Reviewer #1: (No Response)

2. Is the manuscript technically sound, and do the data support the conclusions?

Reviewer #1: Yes

3. Has the statistical analysis been performed appropriately and rigorously? 

Reviewer #1: Yes

4. Have the authors made all data underlying the findings in their manuscript fully available?

Reviewer #1: Yes

5. Is the manuscript presented in an intelligible fashion and written in standard English?

Reviewer #1: Yes

6. Review Comments to the Author

Reviewer #1: The authors have to be clear if the study is prognostic. For instance wording as “protective effect” suggests causality and should be avoided.

7. PLOS authors have the option to publish the peer review history of their article (what does this mean?). If published, this will include your full peer review and any attached files.

Reviewer #1: No

---

## [Editor Report · Acceptance letter]

24 Feb 2022

PONE-D-21-10679R3 

Risk of hospitalization and mortality due to COVID-19 in people with obesity: An
analysis of data from a Brazilian state 

Dear Dr. Reis:

I'm pleased to inform you that your manuscript has been deemed suitable for publication in PLOS ONE. Congratulations! Your manuscript is now with our production department. 

Kind regards, 

on behalf of

Dr. Jennifer A. Hirst 

Academic Editor

PLOS ONE